# Strategic Use of Biodegradable Temporizing Matrix (BTM) in Wound Healing: A Case Series in Asian Patients

**DOI:** 10.3390/jfb15050136

**Published:** 2024-05-18

**Authors:** Angela Chien-Yu Chen, Tsuo-Wu Lin, Ke-Chung Chang, Dun-Hao Chang

**Affiliations:** 1Department of Plastic and Reconstructive Surgery, Chang Gung Memorial Hospital, Linkou Branch, Taoyuan 333, Taiwan; cychen730@cgmh.org.tw; 2Division of Plastic and Aesthetic Surgery, Department of Surgery, Far Eastern Memorial Hospital, New Taipei City 220, Taiwan; twlin@mail.femh.org.tw (T.-W.L.); femh71641@femh.org.tw (K.-C.C.); 3Department of Information Management, Yuan Ze University, Taoyuan 320, Taiwan; 4School of Medicine, National Yang Ming Chiao Tung University, Taipei 112, Taiwan

**Keywords:** biodegradable temporizing matrix, skin substitute, wound healing

## Abstract

Skin and soft tissue reconstruction has long been based on the reconstructive ladder. However, a skin substitute has become popular due to its predictable outcomes, without donor-site morbidity. The biodegradable temporizing matrix (BTM; NovoSorb, PolyNovo Ltd., Port Melbourne, Australia) is a synthetic skin substitute that has recently gained its clinical application. Compared with those of other dermal templates, the clinical efficacy and performance of the BTM are not well established, especially among the Asian population. This study aims to share our experience and strategy of using BTM in various wound conditions. The data of patients who underwent skin and soft tissue reconstruction with BTM at a single institution between January 2022 and December 2023 were reviewed. The patient demographics, wound characteristics, surgical details, secondary procedures, and complications were recorded and analyzed. Postoperative 6-month photographs were collected and independently evaluated by two plastic surgeons and two wound care center nurses using the Manchester Scar Scale (MSS). This study included 37 patients, consisting of 22 males and 15 females with a mean age of 51.8 years (range, 18–86 years old). The wound etiologies included trauma (67.6%), necrotizing soft tissue infection (16.2%), burns (10.8%), toe gangrene (2.7%), and scar excision (2.7%). The average wound area covered by BTM was 50.6 ± 47.6 cm^2^. Among the patients, eight received concomitant flap surgery and BTM implantation, 20 (54.1%) underwent subsequent split-thickness skin grafts (STSG), and 17 had small wounds (mean: 21.6 cm^2^) healed by secondary intention. Infection was the most common complication, affecting six patients (*n* = 6 [16.2%]), five of whom were treated conservatively, and only one required debridement. Thirty-three patients (89.2%) had good BTM take, and only four had BTM failure, requiring further reconstruction. At the last follow-up, 35 out of the 37 patients (94.6%) achieved successful wound closure, and the total MSS score was 10.44 ± 2.94, indicating a satisfactory scar condition. The patients who underwent BTM grafting without STSG had better scar scores than those who received STSG (8.71 ± 2.60 vs. 11.18 ± 2.84, *p* = 0.039). In conclusion, the BTM is effective and feasible in treating various wounds, with relatively low complication rates, and it can thus be considered as an alternative for skin and soft tissue reconstruction. When combined with adipofasical flap reconstruction, it achieves a more comprehensive anatomical restoration.

## 1. Introduction

The concept of the reconstructive ladder was developed to establish priorities for technique selection based on the complexity of the technique and the defect requirements for safe wound closure. However, in the 21st century, modern plastic surgeons are able to weigh the risks and benefits to choose some advanced procedures, such as perforator flap or free flap, to achieve better cosmetic and functional outcomes without compromising safety. However, limitations and challenges are still associated with these reconstructive techniques, especially in regard to donor-site morbidities. Extensive soft tissue defects, characterized by exposed bone and significant scar tissue, obviate traditional skin grafting and jeopardize loco-regional flap reconstruction. While free tissue transfer appears as a potential solution generally, the presence of multiple injuries or patient morbidities might deter individuals from undergoing prolonged flap surgeries, which also necessitate sufficient expertise and training to achieve favorable results. Therefore, it has prompted a paradigm shift toward innovative alternatives, such as artificial skin substitutes. Artificial skin substitute acts as a dermal regeneration template, providing a stable scaffold that promotes the synthesis of neodermal tissues and can protect the wound from infection and moisture loss [1,2].

The NovoSorb biodegradable temporizing matrix (BTM, PolyNovo Ltd., Melbourne, Australia) is a novel and bioabsorbable synthetic polyurethane bilayer dermal template [3,4] recently introduced to Taiwan. The BTM consists of a 2 mm thick bioabsorbable polyurethane open cell foam matrix covered by a non-biodegradable sealing membrane. The open cell matrix permits the infiltration of cellular materials, while serving as a scaffold for the neodermis. Simultaneously, the sealing membrane creates physiological wound closure and also provides a barrier to external bacteria [5]. It was initially indicated for managing pressure ulcers in 2014 [6], being eventually extended to the treatment of full or deep partial-thickness burns, necrotizing fasciitis, surgical and reconstructive wounds, and traumatic wounds [5,7,8].

After debridement, infection control, and wound bed preparation, the reconstruction can initiate with a two-stage process: application of the BTM and delamination of the sealing membrane. The first stage involves the material being filled into a defect, allowing cellular migration, with new blood vessel formation and collagen production, to construct a neodermis. The second stage is performed once the clinician identifies that the tissue is fully integrated throughout the matrix, which normally takes 3–4 weeks. The sealing membrane is gently detached, then removed [7,8]. The biodegradable polyurethane foam gradually degrades by hydrolysis and is fully resorbed into the patient’s body by 12–18 months [9].

Compared with those of other dermal templates, the clinical efficacy and performance of BTM are not well established, especially among the Asian population. The primary purpose of this study is to document our experience of using BTM in various wound conditions. This is one of the largest case series in Asian countries to date, striving to furnish clinicians with a guide for clinical practice.

## 2. Materials and Methods

### 2.1. Study Design and Population of Interest

This monocentric and retrospective study was approved by the Research Ethics Review Committee of Far Eastern Memorial Hospital (FEMH), New Taipei City (No.: 113021-E). The data of patients who underwent skin and soft tissue reconstruction with BTM grafting from January 2022 to December 2023 in FEMH were reviewed. All the operations were performed by the senior author, D.H. Chang. This research aimed to describe the potential effectiveness of BTM in the Asian population.

### 2.2. Clinical Application

The purpose of BTM grafting is divided into two types: graftable and ungraftable wounds. For graftable wounds, the wound can be closed simply by using a skin graft, but BTM is indicated for cosmetic or functional concerns (e.g., the wound is located on the hand or in the joint area). For ungraftable wounds, such as tendon- or bone-exposed wounds, BTM is used to downgrade the reconstruction complexity and avoid donor-site morbidity. The treatment algorithm is illustrated in Figure 1.

The three main stages of wound healing using the BTM were adequate wound debridement with or without negative pressure wound therapy (NPWT) for wound bed preparation and infection control, application of the BTM, and administration of foam dressing (changed every 3 days). Finally, the BTM was delaminated after 4 weeks. The remnant wound was healed by secondary intention or underwent subsequent split-thickness skin grafts (STSG) if the wound size was large or located in a joint area (Figure 2).

The aforementioned procedure is the basic scenario for graftable wounds or tendon/fascia-exposed wounds, which can be promptly covered with the BTM. When a small bone area was exposed, the cortex was first burred or drilled, and the BTM was then utilized. For a bone exposure area that was extensive or involved plate exposure, flap reconstruction was concurrently combined with an overlying BTM graft.

### 2.3. Study Parameters and Evaluator Calibration

All charts, including the surgical records, progress notes, nursing records, and clinical notes were reviewed. The data for the following variables were extracted: patient demographics (age, gender, and comorbidity), wound characteristics (etiology, location, size, and depth), concomitant treatment, secondary procedure, time to wound healing, complications, follow-up period, and pictures taken when returning to the clinic.

The scar condition after BTM application with/without STSG was also evaluated. The clinical photos were obtained 6 months after the operation. The pictures were independently graded by plastic surgeons and by the clinical nurses of the wound care center, using a Manchester Scar Scale (MSS) [10]. The original version of the MSS included five assess parameters: color, shine, contour, distortion, texture, and overall rating with the Visual Analog Scale (VAS). The MSS was also designed for image panel assessment by excluding the texture parameter that was applied in this study. Except for shine (matte: 1, shiny: 2), each parameter was graded from 1 (good scar) to 4 (poor scar). The overall VAS of the scar was rated from 0 (best scar appearance) to 10 (worst scar appearance). The MSS was calculated as the sum of the scores of the four variables and the VAS, ranging from 4 to 24.

### 2.4. Statistical Analyses

For statistical analysis, the Mann–Whitney U test was used to compare the continuous variables in scar evaluation. The significance threshold of the *p*-value was set at 0.05. The analyses were performed using SPSS v. 25 software for Windows (SPSS, Inc., Chicago, IL, USA).

## 3. Results

This study included 37 patients (22 males and 15 females), with a mean age of 51.8 (18–86) years. Among these patients, 25 (67.6%) had trauma-related wounds, six had necrotizing soft tissue infection (16.2%), four had burn injuries (10.8%), one had shock-related toe gangrenes, and one had scar contracture excision. The most common wound location was the leg (51.3%), followed by the foot (27.0%), and the hand (10.8%). Eight patients underwent concomitant flap reconstruction with overlying BTM to cover the bone- or plate-exposed wounds. These flaps included five adipofascial flaps, two muscle flaps, and one free anterior-lateral thigh flap. The patient demographics and wound characteristics are listed in Table 1.

The average wound area covered by BTM was 50.6 ± 47.6 cm^2^. Among the patients, 20 (54.1%), who had a large wound size (mean area: 75.3 cm^2^), required secondary STSG surgery with the average interval of 36.9 ± 10.8 days, and the other 17, with small wounds (mean area: 21.6 cm^2^), were healed by secondary intention.

The outcomes and complications are shown in Table 2. The most common complication was infection (*n* = 6 [16.2%]). The early infection signs usually presented as fluid collection below the silicone sheet. For suspected infection, some slits were made using a blade, and the discharge was squeezed out and sent for culture (Appendix A). Hypochlorous acid solution (Microdacyn^®^ Hydrogel Sonoma Pharmaceuticals, CA, USA) and silver-containing dressing (AQUACEL Ag Foam ConvaTec Inc., North Carolina USA) were used along with oral antibiotics. If the infection could not be suppressed, the silicone sheet would be fully removed, and frequent dressing change would be administered. For the six patients who had BTM infection, five of them were treated conservatively by the abovementioned method, and only one required debridement.

Overall, 33 out of the 37 patients had good BTM take. Only four patients had BTM failure and required further reconstruction, including two additional skin substitutes and two adipofascial flaps. Among the 20 patients who underwent STSG, 17 exhibited a well-taken graft and three exhibited partial graft loss. All of them subsequently healed under wound care.

During the study period, with a mean follow-up time of 7.0 ± 4.9 months, 35 out of the 37 patients (94.6%) achieved wound healing. One patient who did not heal was an 85-year-old lady with a lateral lower leg wound of approximately 40 cm^2^ in size but who refused to undergo STSG. She underwent standard wound care with foam dressing and achieved meaningful wound reduction at the last follow-up (3 months, 6 cm^2^). Another 57-year-old man had shock-related toe gangrene, and he still had a tiny ulcer over his toe stump after treatment. The overall time to wound healing was 87.2 ± 45.3 days, and the patients who underwent BTM grafting without STSG had a shorter time to wound healing (67.2 ± 25.6 days) compared to the other patients.

### 3.1. Scar Appearance Evaluation

The 6-month postoperative photographs of 20 patients were evaluated by two plastic surgeons and two wound care center nurses, and the results are listed in Table 3. The average scores among each category were between 1 and 3, and the total MSS score was 10.44 ± 2.94, indicating a satisfactory postoperative scar condition. The patients who underwent BTM grafting without STSG had better scar scores than those who received STSG only (8.71 ± 2.60 vs. 11.18 ± 2.84, *p* = 0.039). The reason might be the meshed skin graft-related contour change or scar contracture effects.

### 3.2. Case Reports

#### 3.2.1. Case 1 (Figure 2)

An 83-year-old male patient, who had diabetes and chronic obstructive pulmonary disease, suffered from right lower-leg necrotizing soft tissue infection (NSTI). He was hospitalized for antibiotic treatment and underwent serial debridement and NPWT. After the infection had been controlled, a sizable wound measuring 10 × 20 cm^2^, with fibula bone exposure, was left. The BTM implantation was performed and fixed by NPWT. The NPWT was removed 1 week later, and the patient was discharged and used foam dressing coverage. The silicone sheet of BTM was removed after a period of 4 weeks, and well-taken neodermis, with no bone exposure, was noted. He underwent meshed STSG thereafter, and the wound healed uneventfully, with good scar condition at the 6-month follow-up.

#### 3.2.2. Case 2 (Figure 3)

An 80-year-old man, who had diabetes, hypertension, and coronary heart disease, was involved in a traffic accident, causing an open fracture in his left distal tibia. The patient initially underwent debridement, followed by external fixation and NWPT to stabilize the wound milieu. After the wound was stabilized, open reduction internal fixation and wound reconstruction were performed in a combined surgery. Notably, the exposed plate was covered by a posterior tibia artery perforator-based adipofascial turnover flap. The flap’s surface and other residual wounds were all covered by the BTM graft and fixed by NPWT. A secondary STSG was performed 5 weeks later, and the wound healed well during the follow-up.

**Figure 3 jfb-15-00136-f003:**
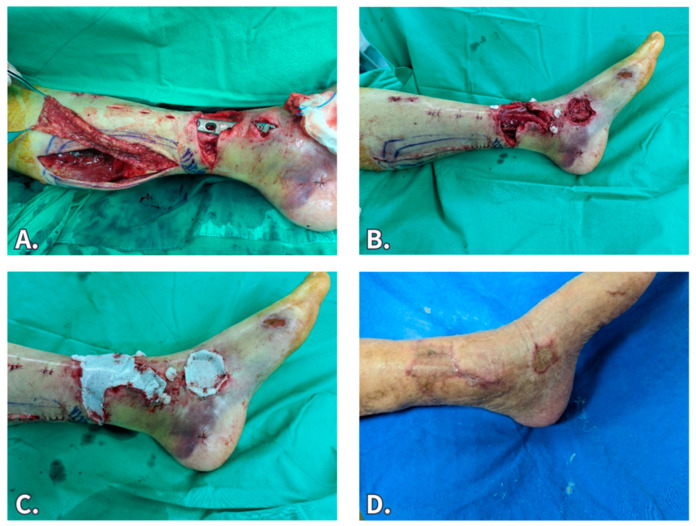
An 80-year-old man had left distal tibia open fracture. (**A**) After undergoing open reduction and internal fixation, the exposed plate was noted, and an adipofascial turn over flap was elevated. (**B**) After the flap was transferred, the plate was covered. (**C**) BTM was used to cover the adipofascial flap. (**D**) Three months after secondary STSG.

#### 3.2.3. Case 3 (Figure 4)

A 66-year-old lady had a right middle-, ring-, and little-finger crush injury that resulted in finger amputation and open fracture. She underwent debridement and bone shortening, and the wounds were wrapped by a BTM graft. The neodermis grew well and STSG was performed 4 weeks later. Despite encountering the setback of partial graft loss at the middle finger, the wound eventually healed, with a favorable outcome under conservative treatment. The final scar condition was satisfactory.

**Figure 4 jfb-15-00136-f004:**
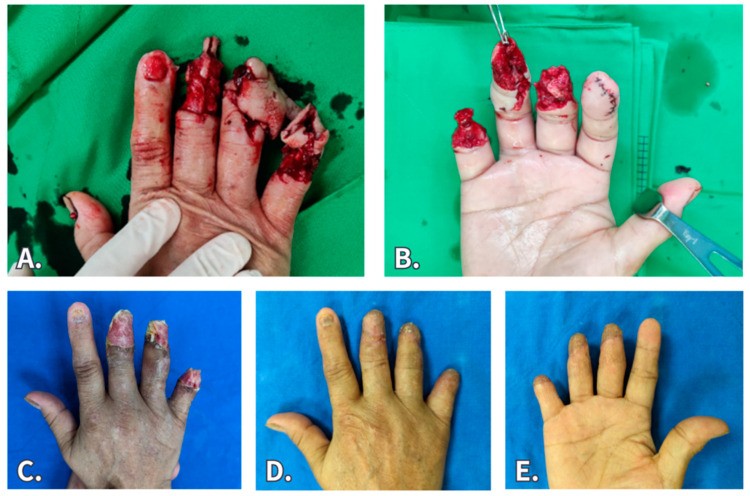
A 66-year-old lady had a right middle-, ring-, and little-finger crush injury. (**A**,**B**) Initial presentation. (**C**) After debridement and bone shortening, the wound was wrapped by BTM. (**D**,**E**) Two months after secondary STSG.

#### 3.2.4. Case 4 (Figure 5)

A 61-year-old man who had diabetes and peripheral artery disease suffered from a right-foot ulcer with NSTI involving the bone and tendon. He underwent serial debridement and percutaneous transluminal angioplasty to improve perfusion. The wound was treated with NPWT for 1 week, and the exposed bone cortex was burred and covered by BTM grafting. However, wound infection developed 2 weeks later, and the wound culture revealed the presence of oxacillin-resistant *Staphylococcus.* Subsequently, the silicone sheet was removed 3 weeks after implantation. Through diligent wound care by daily saline wet dressing and targeted antibiotic treatment (oral linezolid 600 mg Q12h × 7 days), the wound infection was successfully resolved, and most of the neodermis survived. An STSG surgery was conducted thereafter, resulting in the uneventful closure of the wound and subsequent healing.

**Figure 5 jfb-15-00136-f005:**
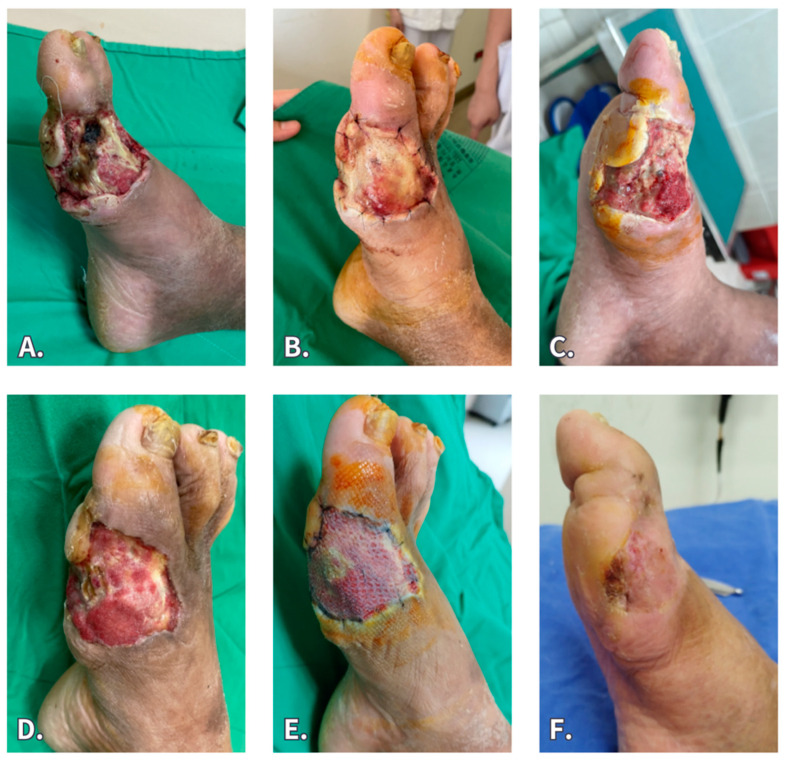
A 61-year-old man had diabetes and peripheral artery disease. (**A**) He suffered from right foot necrotizing soft tissue infection resulting in a wound with bone and tendon exposure. (**B**,**C**) He underwent BTM grafting but wound infection occurred. The silicone sheet was removed at 3 weeks after implantation. (**D**,**E**) After wound care and antibiotic treatment, the wound infection was resolved, and most of the neodermis survived. STSG surgery was then conducted. (**F**) Three months post STSG.

#### 3.2.5. Case 5 (Figure 6)

A 70-year-old man with diabetes and hypertension suffered from a crush injury to his left foot, caused by a 1000 kg iron plate, resulting in first–third toe fracture, fourth and fifth toe traumatic amputation, and skin necrosis. He initially underwent pin fixation, debridement, and full-thickness skin graft at another hospital. However, graft failure with wound necrosis developed, so the patient was transferred to our institution. We performed vigorous debridement, followed by NPWT, after which the wound was covered with BTM grafting. Nevertheless, fluid collection beneath the silicone sheet, with an odor, was noted 2 weeks later, indicating wound infection. Some slits were made by a scalpel on the silicone sheet to drain the fluid. The graft was cared for with hypochlorous gel spray and silver foam dressing, and oral antibiotics were also prescribed. The culture revealed oxacillin-resistant *Staphylococcus*. At 4 weeks post-grafting, the silicone sheet was removed, and the neodermis was nearly fully preserved. Subsequently, he underwent STSG, and the wound healed uneventfully. 

**Figure 6 jfb-15-00136-f006:**
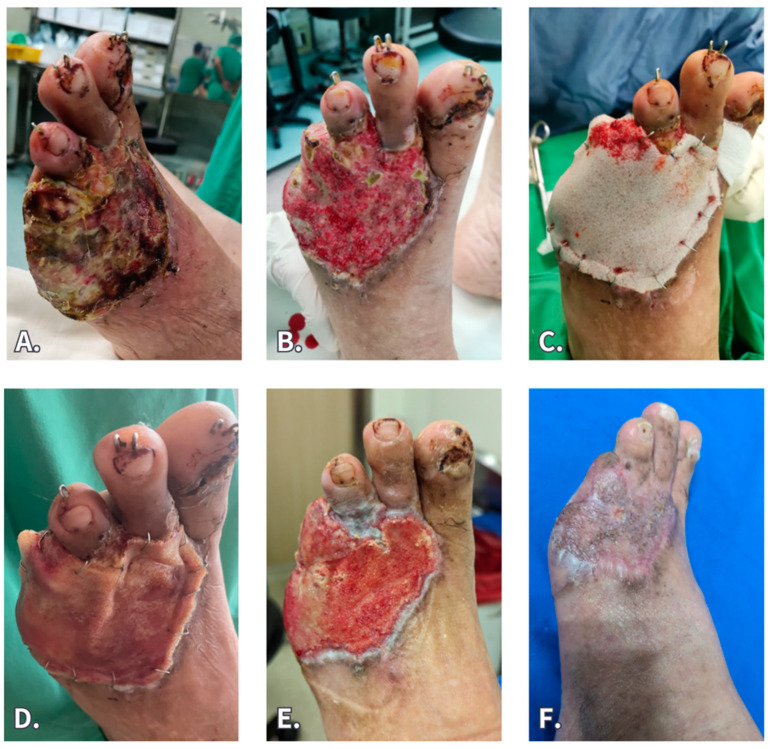
A 70-year-old suffered from left-foot crush injury with 1st–3rd toe phalangeal bone fracture and 4th and 5th toe traumatic amputation. (**A**) On admission, failed full-thickness skin graft (performed at another hospital) and wound infection were noted. (**B**) Serial debridement and NPWT improved the wound bed condition. (**C**) BTM grafting. (**D**) Fluid collection under the silicone sheet with odor was noted 2 weeks after BTM grafting. (**E**). Four weeks after BTM grafting, infection was resolved and the neodermis grew well. (**F**) Two-month follow-up after STSG.

## 4. Discussion

To our knowledge, plenty of commercially available dermal skin substitutes are widely used for skin reconstruction. However, no universal categorization system has been established. Skin substitutes could be classified by their cellularity, the layer to reintegrate (epidermal, dermal, or both), or whether they originate from natural or synthetic sources [2,11]. Our study revealed the efficacy of BTM, a fully synthetic dermal matrix, in wound coverage. Different from biological materials, synthetic substitutes could be synthesized as required and modulated for specific needs, offering precise control over scaffold composition [12]. They could also lower the risks of potential disease transmission or immune rejection, which may arise in xenografts [13,14]. However, biological skin substitutes have a more intact extracellular matrix than synthetic ones, making the neodermis natural because it supports the balance between collagen synthesis and degradation [15]. In addition, the basement membrane in biological matrices facilitates demarcation and cohesion between the epidermal and dermal layers, allowing for re-epithelialization [12,16]. Therefore, it is reasonable to take longer time (e.g., 4 weeks) for a synthetic skin substitute to integrate into the wound bed. Additionally, we also demonstrated that BTM has good healing ability in various wound conditions.

The head-to-head studies comparing biological and synthetic skin substitutes are limited. Integra, a collagen–chondroitin silicone (CCS) bilayer graft (Integra LifeSciences Corp., Princeton, NJ, USA) derived from bovine tissue, has been the most common and widely used biological dermal substitute [17]. Researchers usually use CSS as a comparator to BTM. Cheshire et al. applied BTM and CSS to a nude mice model and revealed that BTM demonstrated a more extensive vascular network but also a greater inflammatory response at 2 weeks post-grafting [12]. Another similar animal study conducted by Banakh et al. [18] also showed that BTM had higher vascularization and fibroblast infiltration and a greater capacity to support human epidermal cells when compared with CSS. Wu et al. had conducted a retrospective clinical study with 97 patients using BTM and CSS. The study showed similar wound closure and complication rates between the two groups, but the BTM group had a lower graft loss rate and required fewer secondary procedures [19]. These studies provided evidence for the potential advantages for the synthetic skin substitutes, although larger-scale clinical studies are required.

BTM is associated with a low infection rate because its synthetic material does not provide nourishment for bacteria [3,20]. According to our study, although infection was the most common complication in the patients who underwent BTM grafting, the overall incidence (16.2%) was acceptably low compared to previous reports [3,19]. Even so, we managed the infection condition well by draining the subseal fluid and discharge and by administering hypochlorous acid gel and silver dressing. Hypochlorous acid products have noncytotoxic properties and a physiological pH, which have been proven to effectively salvage skin graft healing in patients with known bacterial colonization [21]. Combining these methods could protect most of our patients from receiving further debridement surgery, preserving most of the BTM covering the wound. 

Moreover, nearly all the patients in this study underwent NPWT for wound care before BTM grafting, and NPWT was also used as the fixation dressing of BTM if it was suitable. NPWT is superior to conventional dressings because it stimulates granulation tissue formation, decreases blood and serum accumulation, provides a moist environment, encourages vasodilatation, and increases wound bed blood flow, thus improving wound healing [22,23]. Hence, NPWT warrants consideration for enhancing wound bed preparation prior to BTM grafting and optimizing infection control and graft healing after BTM implantation.

For ungraftable wounds, free tissue transfer might be indicated but not always necessary. Skin substitutes with subsequent skin grafting offer an alternative with less donor-site morbidity, simpler procedures, and shorter hospitalization. In addition to split-thickness skin grafting, some of our patients underwent concomitant flap reconstruction, mostly using adipofascial flaps. We preferred to use this flap for lower leg reconstruction especially the distal third defect with implant exposure. Because adipofascial flap can preserve the major leg vessels, with minimal donor-site morbidity and relatively easy and rapid dissection, it has been widely used and reported in previous studies [24,25]. However, the adipofascial tissue might not be a perfect wound bed for STSG take, and graft loss would be sometimes encountered [26]. Therefore, we combined BTM grafting and adipofascial flap to provide a better wound bed with a neodermis. This combination can restore the analogous anatomy of the subcutaneous layer of the skin, in conjunction with the dermal matrix that replaces the dermal layer, offering a reconstruction that is anatomically and aesthetically appealing [27]. BTM was utilized to establish a resilient and flexible tissue layer in such cases. When integrated with skin grafting, this approach yielded better cosmetic results and sustained functional outcomes compared to skin grafting alone [28].

Wagstaff et al. [9], Greenwood et al. [20], and Lo et al. [29] all demonstrated the favorable scarring characteristics and aesthetic outcomes of BTM postoperatively. However, they used a scar evaluation scale different from ours, and their populations of interest were mostly non-Asians. Our study proved that BTM also presented good scarring results in Asian people. The patients who received BTM without STSG had significantly better MSS scores than those who also received STSG, especially the meshed ones. STSG presents many macrophages, mast cells, and fibrocytes, which potentially lead to increased fibrosis and wound contraction [30], contributing to its impact on unsatisfactory scar formation. Based on our experience, we considered secondary healing if the residual defect after BTM coverage was less than 20 cm^2^ in order to achieve preferable outcomes.

The study has several limitations that warrant consideration. First, there is a lack of prospective comparative studies directly comparing the clinical efficacy and outcomes of BTM with other commonly used skin substitutes, such as Integra, which limits the availability of robust evidence regarding BTM’s superiority or equivalence in different wound types and patient populations. Second, the follow-up periods in the current research hinder the assessment of BTM’s long-term durability and performance. Lastly, the absence of patient-reported outcome measures (PROMs) prevents a comprehensive understanding of the patients’ subjective satisfaction levels following BTM reconstruction.

## 5. Conclusions

BTM is effective and feasible in treating various wounds, with relatively low complication rates. It facilitates comprehensive anatomical restoration when combined with adipofascial flap reconstruction, thus yielding superior cosmetic outcomes. In light of its demonstrated efficacy and versatility, BTM emerges as a viable alternative for skin and soft tissue reconstruction within the realm of clinical practice.

## 6. Future Directions

Long-term follow-up: While the current study provides insights into short- to medium-term outcomes, future research should aim to include longer follow-up periods to assess the durability and longevity of BTM-mediated wound healing. Understanding the long-term performance of BTM in terms of scar remodeling, functional outcomes, and recurrence rates would be crucial for informing clinical decision-making.Patient-reported outcomes: Incorporating PROMs in future studies would provide valuable insights into the subjective experiences and satisfaction levels of patients undergoing BTM reconstruction. Evaluating factors such as pain, itching, functional impairment, and overall quality of life would also offer a more comprehensive understanding of the impact of BTM grafting on patients’ well-being.Health economics analysis: Conducting health economics analyses to evaluate the cost-effectiveness of BTM compared to traditional reconstructive techniques and/or other skin substitutes would provide valuable insights for clinicians. Hospitalization costs, operating room utilization, postoperative complications, and long-term resource utilization could be essential in determining the economic impact of adopting BTM in clinical practice.Mechanistic studies: Exploring the cellular and molecular mechanisms underlying BTM-mediated wound healing could provide deeper insights into their mode of action and potential advantages over other skin substitutes.

By addressing these future directions, clinicians can advance our understanding of BTM’s role in skin and soft tissue reconstruction, optimizing its clinical utility for the benefit of patients worldwide.

## Figures and Tables

**Figure 1 jfb-15-00136-f001:**
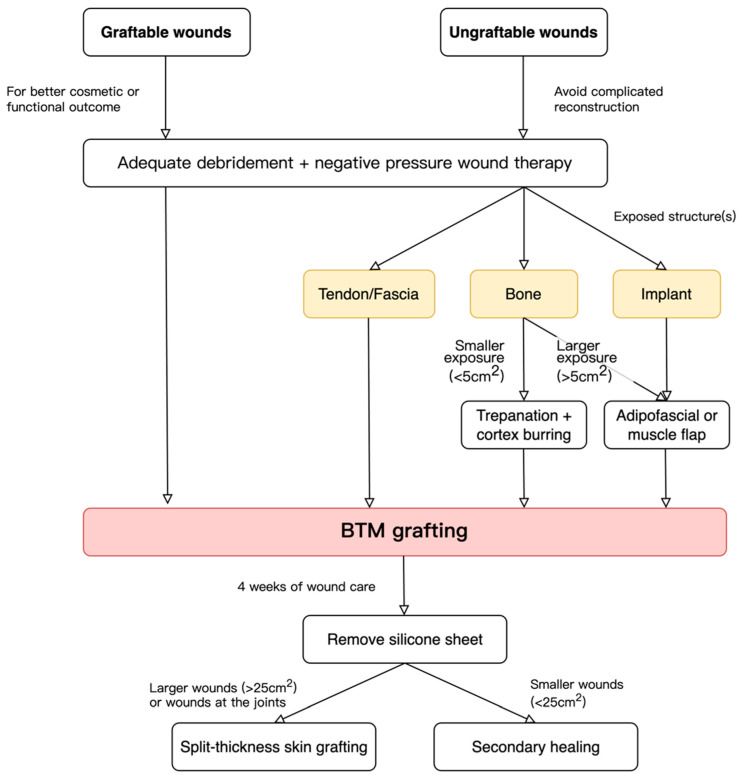
Algorithm for using BTM in skin and soft tissue reconstruction.

**Figure 2 jfb-15-00136-f002:**
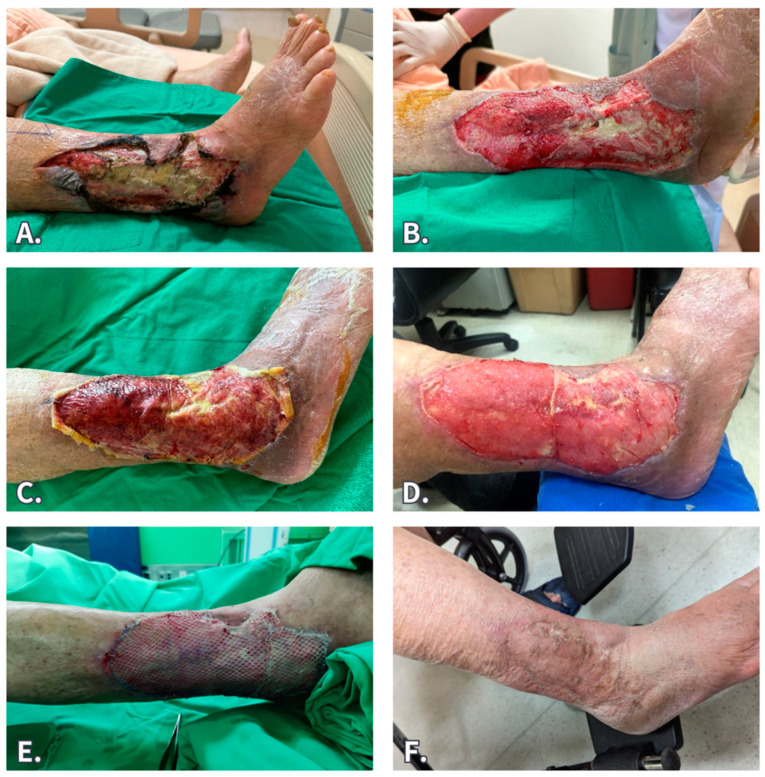
An 83-year-old male patient suffered from right lower leg necrotizing soft tissue infection. (**A**) After first debridement. (**B**) After serial debridement and NPWT, the wound was well granulated, with a small area of fibula bone exposure. (**C**) BTM graft implantation. (**D**) At 4 weeks after BTM grafting, the silicone sheet was removed, and the neodermis growth was good. (**E**) Meshed STSG. (**F**) Six months post STSG.

**Table 1 jfb-15-00136-t001:** Patient demographics and defect characteristics.

	Patients (*N* = 37)	Range/Percentage
**Age (years)**	51.8 ± 21.5	18–86
**Sex**		
**Male**	22	59.5%
**Female**	15	40.5%
**Defect size (cm^2^)**	50.6 ± 47.6	2–180
**Etiology**		
**Trauma**	25	67.6%
**NSTI**	6	16.2%
**Burn**	4	10.8%
**Other**	2	5.4%
**Defect Location**		
**Hand**	4	10.8%
**Arm**	3	8.1%
**Foot**	10	27.0%
**Leg**	19	51.3%
**Trunk**	1	2.3%
**Concomitant procedure**		
**Fracture fixation**	8	21.6%
**Amputation**	5	13.5%
**Flap**	8	21.6%
**Tendon repair**	1	2.3%
**Secondary STSG**	20	54.1%

NSTI: necrotizing soft tissue infection; STSG: split-thickness skin grafting.

**Table 2 jfb-15-00136-t002:** Outcomes and complications of BTM grafting.

	Patients (*N* = 37)	Range/Percentage
**Wound healing**	35	94.6%
**Wound healing time (days) ***		
**Overall (*n* = 35)**	87.2 ± 45.3	20–213
**With STSG (*n* = 20)**	102.3 ± 51.3	53–213
**Without STSG (*n* = 15)**	67.2 ± 25.6	20–108
**BTM complication (*n* = 37)**		
**BTM poor take**	4	10.8%
**Infection**	6	16.2%
**Skin graft complication (*n* = 20)**		
**Skin graft loss**	3	15%
**Infection**	1	5%

* The wound healing time, defined as the duration between BTM grafting and total wound healing.

**Table 3 jfb-15-00136-t003:** Scar evaluation of BTM-grafting patients using Manchester Scar Scale (MSS).

	Overall (*n* = 20)	BTM with STSG (*n* = 14)	BTM without STSG (*n* = 6)	*p* Value
**Color**	2.39 ± 0.55	2.48 ± 0.60	2.17 ± 0.38	0.356
**Shine**	1.33 ± 0.28	1.38 ± 0.27	1.21 ± 0.29	0.154
**Contour**	1.66 ± 0.52	1.79 ± 0.55	1.38 ± 0.31	0.087
**Distortion**	1.93 ± 0.50	2.04 ± 0.45	1.67 ± 0.56	0.072
**VAS**	3.14 ± 1.54	3.50 ± 1.53	2.29 ± 1.29	0.068
**MSS**	10.44 ± 2.94	11.18 ± 2.84	8.71 ± 2.60	**0.039 ***

BTM: biodegradable temporizing matrix; STSG: split-thickness skin grafting; VAS: Visual Analogue Scale; MSS: Manchester Scar Scale; *: statistically significant

## Data Availability

All data generated or analyzed in this study are included in this published article. The datasets used and/or analyzed during the current study are available from the corresponding author.

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
