# Peer review of "Strategic Use of Biodegradable Temporizing Matrix (BTM) in Wound Healing: A Case Series in Asian Patients"

_jfb, 2024, doi:10.3390/jfb15050136_

Round 1
Reviewer 1 Report
Comments and Suggestions for Authors
Dear authors,
Congratulations on your retrospective study, which opens the way for Biodegradable Temporizing Matrix use in wound healing.
Introduction - Do you believe the reconstructive ladder is deeply rooted in modern plastic surgery? Many colleagues favor an elevator reconstructive concept, that allows you to select the first best option for the case, not merely the first and simplest one. Maybe you can rephrase it.
Do you also belive that Asian patients do differ from other groups and if so in what exactly (healing, tissues?)
Fig 1 - can you elaborate on the meaning of large versus small? is there a definite dimension for this?
Results are presented clearly.
Discussion section - you compare this product with other skin substitutes, and that's fine. I would appreciate a small section on this study's limitations, as you showed future directions.
Reference 8 - I do believe that it should be rewritten - Experience with <scp>NovoSorb</scp>® Biodegradable Temporising Matrix in reconstruction of complex wounds.
Comments on the Quality of English LanguageMinor English editing should add to the value of the article, also this could reflect the author's originality (use of tenses, phrasing).
Eg. The first stage involves the material being filled 72
into a defect, allowing cellular migration with new blood vessel formation and collage
production to construct a neodermis. The second stage is performed once the clinicia
identifies the tissue is fully integrated throughout the matrix, the sealing membrane
would be gently detached then removed
Reviewer 2 Report
Comments and Suggestions for Authors
Chen et al.'s work is highly significant for wound care management. The authors provide a comprehensive description of using BTM for both graftable and non-graftable wounds. They engage in a thoughtful discussion, effectively highlighting current market gaps and outcomes.
I have a few minor comments:
Lines 50-51: The concept needs further elaboration.
Lines 71-75: Could the authors discuss the typical timing (in clinics) of BTM application?
Table 2 would benefit from clarification of the initial section; it's unclear what the numbers represent for wound healing time. Are the days indicated for both with and without STSG?
Lines 173-177: What treatment was administered to the 85-year-old patient who declined STSG? What was the outcome?
Line 234: What treatment was provided to the 61-year-old man for the infection? This detail was included for other patients but not for this one.
In summary, what are the potential drawbacks or limitations of their study?
